# A Novel Solution to Find the Dynamic Response of an Euler–Bernoulli Beam Fitted with Intraspan TMDs under Poisson Type Loading

**Alberto Di Matteo [1], Iain Peter Dunn [1,*], Giuseppe Failla [2] and Antonina Pirrotta [1]**

[1]    Dipartimento di Ingegneria, Università degli Studi di Palermo, 90128 Palermo, Italy; alberto.dimatteo@unipa.it (A.D.M.); antonina.pirrotta@unipa.it (A.P.)

[2]    Dipartimento di Ingegneria Civile, dell'Energia, dell'Ambiente e dei Materiali, Università degli Studi Mediterranea di Reggio Calabria, 89124 Reggio Calabria, Italy; giuseppe.failla@unirc.it

*    Correspondence: iainpeter.dunn@unipa.it; Tel.: +39-3911115679

**Abstract:** This contribution considers a virtual experiment on the vibrational response of rail and road bridges equipped with smart devices in the form of damping elements to mitigate vibrations. The internal damping of the bridge is considered a discontinuity that contain a dashpot. Exact complex eigenvalues and eigenfunctions are derived from a characteristic equation built as the determinant of a $4 \times 4$ matrix; this is accomplished through the use of the theory of generalized functions to find the response variables at the positions of the damping elements. To relate this to real world applications, the response of a bridge under Poisson type white noise is evaluated; this is similar to traffic loading that would be seen in a bridge's service life. The contribution also discusses the importance of smart damping and dampers to sustainability efforts through the reduction of required materials, and it discusses the role played by robust mathematical modelling in the design phase.

**Keywords:** Euler Bernoulli beam; poissonian loading; tuned mass damper

---

## 1. Introduction

Modelling and simulation are becoming increasingly important enablers for the analysis and design of complex systems. In the domains of structural design, automotive design, mechanical design, biomechanics, and transport infrastructure, the notion of a "virtual experiment" is a valuable tool for checking and optimizing extensively the complex designs before a full scale realization is ever made.

Analytical models represent a valuable tool to understand the dynamic response of currently existing transport infrastructure, namely rail and road bridges, as well as those currently being designed.

In rail networks, small imperfections that arise over time caused by wear, subsidence, and a lack of proper maintenance can cause significant problems to arise if left unchecked, leading to misalignment, the tilting of sleepers, and gauge imperfections [1].

Of course, railway networks are regularly maintained to avoid the most serious cases, but the elimination of all forms of imperfection is, for all intents and purposes, impossible. As such, these cases of small imperfections provide a significant source of excitation [2].

As such, the ability to understand the limitations of both existing structures and those in design is of paramount importance to understand and then mitigate the effects of this excitation.

The dynamic analysis of structures is a particularly well studied field that has been extensively researched in recent years, but this was not always the case. A number of engineering disasters forced design standards to be updated and made more robust, while, at the same time, advances in computer technology have allowed increasingly complex problems to be modelled and solved, thus facilitating

this robust analysis. Significant advances in the transport industry, such as automation and the increasing interest in high speed rail projects, have again highlighted the importance of dynamic analysis in structures that can be typically modelled as discontinuous Euler–Bernoulli beams i.e., rail or road bridges.

By analyzing the effects of moving loads that are used to simulate traffic loading under which these structures would be subjected, a greater understanding of the dynamic response that we can expect to see in real world applications can be obtained. This type of forcing action is used because bridges subjected to moving loads have greater maximum deflections and maximum bending moments than bridges that are subjected to static loads. Furthermore, in real world applications, bridges are typically only subjected to moving loads because it is rare that vehicles remain static on them for any significant period of time.

Due to the aforementioned advances in rail technology, bridges are likely to be subjected to ever increasing dynamic forces as increasingly fast locomotives utilize existing infrastructure, and so this analysis takes on a greater importance due to the danger of a locomotive reaching a beam's critical velocity; at this velocity, serious damage to the beam can be sustained.

A potential barrier standing in the way of the implementation of high-speed rail projects, which are seen as significantly important to reducing carbon emissions by functioning as viable alternatives to short haul intercity flights, is the aging of infrastructure, a process that is occurring all over Europe. A thorough analysis of current infrastructure combined with the addition of adequate damping to mitigate the effects of the increased forces could reduce costs of these projects by removing the need for renewing and rebuilding existing infrastructure.

Generally, the structural behavior of bridges under dynamic loading can be described by resorting to the well-known Euler–Bernoulli beam model. In this regard, a number of contributions can be found in the literature, showing how this model effectively takes the main features of bridge dynamics into account [3].

The Euler–Bernoulli model has been applied in multiple studies to great effect. Law and Zhu [4] developed damage model in which the Euler–Bernoulli beam represented an existing bridge model. Experimental tests were conducted on a reinforced concrete bridge to validate the proposed analytical method, and this was considered "accurate enough to assess the crack damage in the concrete bridge" [4]. Zou et al. [5].then conducted a comparative study of various numerical models to consider the effects of a moving load crossing a bridge. This study used benchmark testing and results from previous studies on existing bridges to validate their model, and they concluded that the Euler–Bernoulli model was sufficiently accurate.

It should be stressed that, typically, a dynamic load will not directly cause a major structural failure; however, continuous loading can lead to the degradation of a bridge, and this can, in turn, lead to bridge failure [6].

Many studies have utilized data obtained by operational modal analysis (OMA) to compare the response of existing infrastructure with analytical models [3–5,7,8].and these studies have generally managed to obtain very good correlations between both data sets.

While each of these studies has utilized the Euler–Bernoulli model to accurately model the response of existing bridges, they also used the classical analytical model. The analytical model that is presented in this paper is far less computationally demanding than the classical model, but it still provides exact results. While most current studies focus on deterministic solutions for known loadings, a number of studies have considered the effect that series of random moving loads have on bridges subjected to them. In many studies concerning the stochastic response of Euler–Bernoulli beams, the random loading that is applied follows a Poissonian distribution, "constituted by a train of impulses of random amplitudes occurring at random time instants" [9].thus giving a stochastic output due to these random elements. Traffic loading (particularly considering road traffic, although this is also applicable to rail traffic) can be described as a Poissonian process because the magnitude of the forces is random, as are their arrival times [10].

Ricciardi [11] proposed a method of modelling the forcing action as a filtered Poisson process; this is obtained by finding "the response of a linear undamped oscillator excited by a Poisson white noise process."

The dynamic response of a Euler–Bernoulli beam can only be accurately obtained by a very small number of methods, the most common ones being computer models such as the finite element method (FEM) and the classical numerical method. However, these methods each have significant drawbacks: the FEM is accurate at lower modes, but as the number of modes under consideration increases, so too does the percentage error in the eigenvalues and eigenfunctions obtained. This is also true as the number of spans increases, so a beam with a high number of spans cannot be as effectively modelled by the FEM. The classical numerical method (CM) does not have this limitation, as it provides exact eigenvalues and eigenfunctions regardless of the number of spans or the number of modes considered; the drawback with this method, however, is the large amount of computational time required to calculate the solution, because, as the number of spans increases, the complexity of the matrices contained in the solution of the CM grows and the required computing power exponentially increases.

This paper expands on the work conducted by Adam et al. [12] in which a novel numerical method was developed to find the response to a series of moving loads acting on a Euler–Bernoulli beam equipped with tuned mass dampers; this method can consider, as the CM does, complex eigenfunctions caused by damping elements, in this case from the tuned mass dampers, where the damping is localized at a specific point rather than proportionally distributed throughout the entire length of the beam, as is often assumed. The method is then extended to consider a series of moving loads following a Poisson type distribution where they have random magnitudes and random arrival times, and the obtained results should, therefore, more closely reflect the response of a road bridge subjected to normal traffic loading.

The SMARTI (sustainable, multifunctional, automated, resilient, transport infrastructure) Project concerns a wide variety of novel ideas that aim to improve the design of future and existing transport infrastructure, in that regard, this paper aims to address the sustainability and resiliency pillars of the SMARTI vision that is designed to serve as a guideline to achieve sustainable, multifunctional, automated, and resilient transport infrastructure. Often, these pillars are interlinked, and nowhere is this clearer than in the design phase, where the development of sustainable architecture must consider resiliency and where automation and multifunctionality are applied to reach sustainability and resiliency goals. Current approaches used in the design of "smart" road bridges are discussed, and the importance of robust, efficient, and highly accurate analytical application and validation of these methods is stressed. This is done with a view to achieve a greater understanding of bridge dynamics that can be used in the design phase to ensure that future rail and road infrastructure is more sustainable by not only reducing the material requirements in the construction phase but by also facilitating more robust structures that require less costly maintenance and that have a longer service life.

## 1.1. Smart Dampers

The SMARTI pillars covered by this project are diverse, and they principally concern sustainability through the use of novel materials and applications using dampers. Our main area of focus is the mathematical modelling of a beam's response when fitted with damping devices. The use of such devices has been extensively studied and has shown that structures in which they are equipped can be considered more sustainable and resilient due to the reduced requirement of materials and maintenance [13].

There are two main principles governing the vibration control of structures: passive control and active control. "Passive control devices are systems which do not require any external power source, imparting forces that are developed in response to the motion of the structure" [14]. Active control devices "rely on a control algorithm" [15] and thus require an external power source unless the power required to run the control algorithm can be generated by the damper itself.

This idea of generating power from the damping action gives rise to the idea of a new breed of smart dampers. Smart dampers are not a new idea, and they come in many different forms, including tuned mass dampers (TMDs) containing some form of a control system that allows for a more effective reduction of maximum displacement, dampers capable of energy harvesting, or even just damping systems containing sensors that provide real-time information on the operating state of the structures to which they are applied. Combining the real time sensors and active control in a damper that generates its own power could allow for significant advances in the utilization of such damping devices.

The most prominently researched "smart-damper" is the semi-adaptive tuned mass damper, which is a TMD that contains either a variable stiffness element or a variable damping element, as well as some form of control system to regulate these properties.

Damping elements with variable damping coefficients take many forms: one prominent type is the magnetorheological (MR) damper, which is a damper in which the fluid has an MR property. An MR fluid is one in which the rheological properties are altered in the presence of a magnetic field [16], thus allowing the viscosity and friction coefficient to be altered through the use of a control system where the current passing through an electromagnet located in the piston controls the amount of resistance and, hence, the damping coefficient. As with all viscous dampers, however, these properties are also highly dependent on the temperature of the fluid, which causes problems in the production of an accurate mathematical model for real world applications because the effect of temperature is considered to increase exponentially [17].

Variable stiffness elements are also common in active and semi-active TMDs. Commonly, an electromechanical actuator is used in conjunction with an arrangement of multiple springs, and the control system can raise or lower the effective stiffness by changing the orientation of the springs [18].

Hybrid dampers also fit within the topic of smart dampers. Hybrid damping systems most commonly combine base isolation in the form of seismic isolators with traditional damping technologies [19]. These hybrid dampers can make use of active or passive damping, and their use is currently gaining scientific attention.

Smart dampers also take the form of multifunctional dampers wherein the damping or the stiffness can be used to harvest energy and convert it to useful electrical power. This can be achieved with electromagnetic dampers. Shen and Zhu [20] discussed this type of damper, presenting a novel form of electromagnetic damper, optimized for energy harvesting. This damper harvests energy by absorbing the vibrations of a system and converting it into usable electrical power, a process that is achieved via electromagnetic induction [20]. This idea, however, is far from new, and multiple studies have been devoted to the idea of harvesting energy that would otherwise be lost by using novel damping devices. It was the principle of piezoelectric energy harvesting that inspired these studies, and, on a smaller scale, piezoelectric energy harvesting has been implemented as a method of generating electricity from ambient vibrations and in damping systems [21], Piezoelectric energy harvesting works by a similar process but on a much smaller scale. Piezoelectric materials generate electric potential when subjected to mechanical stress, though this potential is very small; however, connecting multiple piezoelectric materials in series allows usable power to be extracted [22]. Lafarge et al. [21] discussed the use of these devices in a motor vehicle's suspension system, but there is no reason that this could not be scaled up to larger structures.

The tuned liquid column damper (TLCD) could be considered as another type of multifunctional "smart damper." TLCDs are passive damping devices that make use of a liquid contained within a U-shaped column to mitigate vibrations in structures [23], TLCDs have a performance similar to that of TMDs but with the added advantage of a large source of water that can be put to a second use such as firefighting [24]. Further, TLCDs are highly cost effective and low maintenance, making them a very attractive option as a substitute for a TMD.

The final type of smart damper that is discussed in this study is an adaptive TMD, which can optimize its position depending on the type of loading applied and the beam response. This innovative TMD was discussed in Soroush et al.'s study [25], and it is based on work by Zhou and Sun [26],

in which an MR damper for use in these applications was studied and experimentally validated. This TMD is not just adaptive due to its ability to optimize its position but also due to its variable damping coefficient, which is similar to the operation of an MR damper.

All of these studies on a damper capable of optimizing its position, however, concerned the reduction of damping on stay cables of suspension bridges and not on an actual bridge itself. There are a number of reasons for this, but, simply put, there seems to be little advantage to using these types of devices on more substantial structures because the energy required to move far larger TMDs and the vibrations caused by these movements seem to negate any positive effects that could be gained.

To apply any of these smart dampers to the method proposed below, only a few parameters must be understood. Assuming that they will take the form of a TMD, the stiffness, damping coefficient, and lumped mass must be known. It must also be assumed that an exact mechanical value can be attributed to each of these parameters, regardless of the way the stiffness or damping is created.

With variable damping or stiffness, however, problems arise in any modelling method because previously constant values are now variable. The major advantage of this, however, is that the control system will likely calculate the variable attributes of the TMD based on the frequency of the vibrating beam, as is often the case currently [18]. In the proposed method, the frequency-dependent stiffness and the frequency-dependent damping are already used through a Fourier transform of the standard spring and damper calculations; therefore, adaptation to a new equation for calculating the frequency-dependent variable stiffness and variable damping coefficient should be relatively straightforward. In the case of a damper capable of optimizing its position, the application to the method proposed in this paper is far less straightforward.

The greatest barrier to the incorporation of these types of devices is a lack of faith on the part of engineers and legislative bodies. As with all novel technologies, there is a level of resistance preventing them from becoming widespread, but this resistance is not without merit. The use of best practices and tried and proven methods is often more prudent than novel, unproven methods. This is where the mathematical modelling presented in the following sections may prove useful. By providing a simple-to-use and accurate method for predicting the response of a rail or road bridge when subjected to loading, it is the authors' ambition to provide a weight of scientific evidence supporting the use of novel materials and devices in these applications so that their use may become more diffused in the coming years.

### 1.2. Tuned Mass Dampers in Existing Rail and Road Infrastructure

The TMD has been applied to multiple bridge structures, from small pedestrian foot bridges to large rail bridges. They are employed for a number of reasons, from passenger comfort, as a bridge that vibrates with a high mean displacement does not inspire confidence even when perfectly safe. Secondly, they are installed to increase the service life of bridges by lowering fatigue, and they are also employed to allow for the passage of "live loads" [27], i.e., moving loads that induce vibrations due to their high speed; as such, a number of existing bridges have been modified throughout Europe to conform to new Eurocodes because "existing bridges must carry the traffic category requested for new bridges" [28].

Nugroho et al. [29] performed a particularly interesting analysis of an existing rail bridge. The bridge in question, the 244.3 m long Cisomang Railway Bridge in Indonesia, does not currently have a TMD attached, and the study aimed to determine to what extent the bridge's dynamic response could be improved by the addition of a TMD. The bridge meets all current Indonesian building standards, but it falls short of meeting the Eurocode EN 1990:2002 [30] standard for any speed over 50 km/h. The study produced a detailed model of the Cisomang Railway Bridge and compared the dynamic response with and without a TMD, ultimately finding that the application of one TMD with a mass ratio of 1.6% would improve the safety of the structure and allow it to conform to more stringent safety and comfort criteria.

## 2. Problem Statement

One of the simplest cases of a discontinuous beam is a two-span beam fitted with one TMD at the midpoint. This allows for the description of the proposed method without the introduction of too many terms, thus retaining clarity. It should be emphasized, however, that when using the method proposed in this paper, any number of TMDs could be fitted at any point in the beam's span without changing the steps required to find the equation of motion (EoM). This will become clear.

When using the proposed formulation, supports, lumped masses, TMDs, and, indeed, any other attachment that does not cause localized rotation can be modelled as a shear discontinuity acting on a specific point. Shear discontinuities could otherwise be described as point forces located at the point of attachment of the TMD, which allows the EoM to take the form [12]:

$$EI\frac{\partial^4 w(x,t)}{\partial x^4} + \overline{m}\frac{\partial^2 w(x,t)}{\partial t^2} + R(x,t) = f(x,t),\tag{1}$$

where "tilde" means generalized derivative, $EI$ is the flexural rigidity, $\overline{m}$ is the mass per unit length, $w(x,t)$ is the dynamic response of the beam in terms of the transversal displacement in the space and time domains, $R(x,t)$ is a generalized function [31] used to account for the discontinuities present (in this case, associated with $N$ TMDs) and $f(x,t)$ is the forcing action. As the generalized function representing the TMDs [32], $R(x,t)$ takes the form:

$$R(x,t) = -\sum_{j=1}^{N} P_j(t)\,\delta(x - x_j),\tag{2}$$

where $P_j(t)$ is the shear reaction force—a point load—and $\delta(x - x_j)$ is a Dirac's delta function positioning the reactionary force at $x_j$, the location of the TMD.

*Free Vibration*

In free vibration, the left-hand side of the EoM is equal to zero, and the EoM takes the form:

$$EI\frac{\partial^4 w(x,t)}{\partial x^4} + \overline{m}\frac{\partial^2 w(x,t)}{\partial t^2} + R(x,t) = 0.\tag{3}$$

At this point, the separable variables approach is applied, wherein the time and space domains are considered independently:

$$w(x,t) = \psi(x)g(t),\tag{4}$$

where $g(t)$ can also be defined as: $g(t) = e^{i\omega t}$.

The space domain term $\psi(x)$ and the time domain term $g(t)$ are split, thus allowing for the solution of the beam's eigenvalues and eigenfunctions; however, the reactionary forces cannot be handled in this manner because the point force caused by the discontinuity is in the time domain only. $R(x,t)$ could easily be transferred into just the time and space domains by splitting $P_j(t)$ from $\delta(x - x_j)$. To retain the forcing term, $P_j(t)$ must be Fourier transformed [33], thus leading to the frequency-dependent term:

$$P(x,\omega) = -\sum_{j=1}^{N} \varphi_j(\omega)\delta(x - x_j),\tag{5}$$

where $\delta(x - x_j)$ is a Dirac's delta function that specifies the application point of the force as before, and [34]:

$$\varphi_j(\omega) = -K_{TMD\ j}(\omega)\,\psi(x_j),\tag{6}$$

where $\psi(x_j)$ is the eigenfunction of deflection at the point $x_j$ and $K_{TMD\ j}(\omega)$ is the frequency-dependent stiffness of the TMD given by the Fourier transform of the term in the time domain [35]:

$$K_{\text{TMD}\ j}(\omega) = \frac{\left(k_{TMD\ j}+i\omega\ c_{TMD\ j}\right)\text{M}_{TMD\ j}\ \omega^2}{\text{M}_{TMD\ j}\ \omega^2 - \left(k_{TMD\ j}+i\omega\ c_{TMD\ j}\right)}, \tag{7}$$

where $\text{M}_{TMD\ j}$ is the magnitude of the lumped mass of the $j$th TMD, $k_{TMD\ j}$ is the spring stiffness of the $j$th TMD, $c_{TMD\ j}$ is the damping coefficient of the dashpot that forms part of the $j$th TMD, and $\omega$ is the frequency.

## 3. Eigensolution

The exact modes of vibration can be found by applying the separable variables method, Equation (4), allowing for the transversal displacement $w(x,t)$, rotation $\theta(x,t)$, bending moment $m(x,t)$, and shear $Q(x,t)$ to be expressed in dimensionless form as:

$$w(x,t) = \psi(x)e^{i\omega t}\ ;\ \theta(x,t) = \vartheta(x)e^{i\omega t}, \tag{8}$$

and

$$m(x,t) = \mu(x)e^{i\omega t}\ ;\ Q(x,t) = \chi(x)e^{i\omega t}, \tag{9}$$

This gives the four eigenfunctions of the response variables $\psi(x)$ deflection, $\vartheta(x)$ rotation, $\mu(x)$ bending moment, and $\chi(x)$ shear. These eigenfunctions are related through a derivative method where:

$$\begin{aligned}
\vartheta(x) &= \frac{\widetilde{d\psi(x)}}{dx}; \\
\mu(x) &= -\frac{\widetilde{d\vartheta(x)}}{dx}; \\
\chi(x) &= \frac{\widetilde{d\mu(x)}}{dx}; \\
\frac{\widetilde{d\chi(x)}}{dx} &+ \sum_{j=1}^{N} \varphi_j(\omega)\ \delta(x-x_j) + \sigma^2\psi(x) = 0.
\end{aligned} \tag{10}$$

From these relations, the free vibration of the beam in the space domain can then be expressed as:

$$\frac{\widetilde{d^4}\psi(x)}{dx^4} + P(x,\omega) - \sigma^2\psi(x) = 0 \tag{11}$$

where $\sigma^2 = \left(\omega^2\ \overline{m}L^4\right)/EI$.

Equation (7) shows that the reaction force of the TMD attached at point $x_j$ depends on the frequency-dependent term concerning the spring stiffness, the damping coefficient, and the attached mass.

In Equation (5), the term $\varphi_j(\omega)$ representing the frequency-dependent stiffness based on the TMD's parameters and the deflection at $x_j$ is an unknown due to the presence of the eigenfunction of deflection. Therefore, the matrix approach must be applied. In this regard, let $\mathbf{Y}(x)$ be the vector built from the response variables of the eigenfunctions:

$$\mathbf{Y}(x) = \begin{bmatrix} \psi(x) & \vartheta(x) & \mu(x) & \chi(x) \end{bmatrix}^T, \tag{12}$$

Failla [36] proposed a method to find the unknown $\varphi_j(\omega)$ in a closed form. Through the solution of a linear system involving the $4 \times 1$ vector $\mathbf{c}$, which is constructed from the four integration constants that are found from the solution of the homogeneous equation associated with Equation (11), a closed form expression can be obtained. This leads to the following closed analytical expression of $\mathbf{Y}(x)$:

$$\mathbf{Y}(x) = \widetilde{\mathbf{Y}}(x)\mathbf{c}, \tag{13}$$

where $\widetilde{\mathbf{Y}}(x)$ is a $4 \times 4$ matrix given by the following expression:

$$\widetilde{\mathbf{Y}}(x) = \mathbf{\Omega}(x) + \sum_{j=1}^{N} \mathbf{J}\left(x, x_j\right) \varphi_j(\omega) \tag{14}$$

in which each row represents an individual eigenfunction: deflection, rotation, bending moment, and shear, respectively, for rows 1–4

$$\mathbf{\Omega}(x) = \begin{bmatrix} \Omega_{\psi_1} & \Omega_{\psi_2} & \Omega_{\psi_3} & \Omega_{\psi_4} \\ \Omega_{\vartheta_1} & \Omega_{\vartheta_2} & \Omega_{\vartheta_3} & \Omega_{\vartheta_4} \\ \Omega_{\mu_1} & \Omega_{\mu_2} & \Omega_{\mu_3} & \Omega_{\mu_4} \\ \Omega_{\chi_1} & \Omega_{\chi_2} & \Omega_{\chi_3} & \Omega_{\chi_4} \end{bmatrix}, \tag{15}$$

In the interest of clarity, the matrix $\mathbf{\Omega}(x)$ is expanded below to show the terms contained in the general solution of the homogeneous equation and the derivative method used to relate the eigenfunctions throughout the subsequent rows:

$$\begin{array}{llll} \Omega_{\psi_1}(x) = e^{-\sigma x} & \Omega_{\psi_2}(x) = e^{\sigma x} & \Omega_{\psi_3}(x) = \cos(\sigma x) & \Omega_{\psi_4}(x) = \sin(\sigma x) \\ \Omega_{\vartheta_1}(x) = -\sigma e^{-\sigma x} & \Omega_{\vartheta_2}(x) = \sigma e^{\sigma x} & \Omega_{\vartheta_3}(x) = -\sigma \sin(\sigma x) & \Omega_{\vartheta_4}(x) = \sigma \cos(\sigma x) \\ \Omega_{\mu_1}(x) = -\sigma^2 e^{-\sigma x} & \Omega_{\mu_2}(x) = -\sigma^2 e^{\sigma x} & \Omega_{\mu_3}(x) = \sigma^2 \cos(\sigma x) & \Omega_{\mu_4}(x) = \sigma^2 \sin(\sigma x) \\ \Omega_{\chi_1}(x) = \sigma^3 e^{-\sigma x} & \Omega_{\chi_2}(x) = -\sigma^3 e^{\sigma x} & \Omega_{\chi_3}(x) = -\sigma^3 \sin(\sigma x) & \Omega_{\chi_4}(x) = \sigma^3 \cos(\sigma x) \end{array} . \tag{16}$$

The discontinuities are considered (as shown in Equation (14)) through the vector $\mathbf{J}\left(x, x_j\right)$ which can be expanded as [37]:

$$\mathbf{J}\left(x, x_j\right) = \left[ J_{\psi}{}^{(p)} J_{\vartheta}{}^{(p)} J_{\mu}{}^{(p)} J_{\chi}{}^{(p)} \right]^{T}. \tag{17}$$

This vector is constructed of the particular integrals associated with a TMD for each of the four response variables, and they are related through the same derivative method as before.

At this point, the boundary conditions at the extremes of the beam are enforced:

$$\mathbf{B} \, \mathbf{c} = 0, \tag{18}$$

where $\mathbf{B}$ is a $4 \times 4$ matrix constructed from enforcing the boundary conditions on the matrix $\widetilde{\mathbf{Y}}(x)$, and, as before, $\mathbf{c}$ is a $4 \times 1$ vector of the unknown constants.

From here, the characteristic equation can then be built as the determinant of the $4 \times 4$ matrix $\mathbf{B}$:

$$\det(\mathbf{B}) = 0. \tag{19}$$

Then, the non-trivial solutions of $\mathbf{c}$ are found, and exact closed form expressions can be built for the beam's eigenfunctions. Due to the presence of a dashpot in the TMD model, localized damping will be present, and, therefore, the eigenfunctions and the mode shapes will be complex.

## 4. Orthogonality Conditions

Following the procedure presented in [38], the orthogonality conditions are built to derive the particular impulse response function of this beam.

Firstly, the EoM in free vibration in the form shown below is considered:

$$\frac{\widetilde{d^4}\psi_m(x)}{dx^4} - \sigma_m{}^2 \psi_m(x) + \sum_{j=1}^{N} K_{TMD\ j}(\omega_m)\ \psi_m\left(x_j\right) = 0 \tag{20}$$

Next, multiplying the EoM at mode $m$ by $\psi_n(x)$ and at mode $n$ by $\psi_m(x)$ and then integrating between 0 and L with respect to x:

$$\int_0^L \frac{\widetilde{d^2}\psi_m(x)}{dx^2}\frac{\widetilde{d^2}\psi_n(x)}{dx^2}\,dx - \sigma_m{}^2 \int_0^L \psi_{mn}(x)dx + \sum_{j=1}^N K_{TMD\,j}(\omega_m)\psi_{mn}(x_j) = 0 \qquad (21)$$

and

$$\int_0^L \frac{\widetilde{d^2}\psi_n(x)}{dx^2}\frac{\widetilde{d^2}\psi_m(x)}{dx^2}\,dx - \sigma_n{}^2 \int_0^L \psi_{mn}(x)dx + \sum_{j=1}^N K_{TMD\,j}(\omega_n)\psi_{mn}(x_j) = 0 \qquad (22)$$

where $\psi_{mn}(x) = \psi_m(x)\,\psi_n(x)$.

Integrating by parts and subtracting Equation (22) from Equation (21) then yields the first orthogonality condition:

$$\left(\sigma_m{}^2 - \sigma_n{}^2\right)\int_0^L \psi_{nm}(x)dx + \sum_{j=1}^N \left[K_{TMD\,j}(\omega_n) - K_{TMD\,j}(\omega_m)\right]\psi_{nm}(x_j) = 0. \qquad (23)$$

The second orthogonality condition is then found by multiplying Equation (21) by $\sigma_n$ and Equation (22) by $\sigma_m$ and then subtracting Equation (22) from Equation (21).

## 5. Forced Vibrations

These orthogonality conditions are then used to derive the beam's response to arbitrary loading. This is accomplished by using the complex modal superposition principle, as defined by [10], where the complex modal impulse response function is used. This leads to [2].

$$w(x,t) = \sum_{k=1}^{\infty} \psi_k(x)\,\frac{1}{i\,\Xi_k\,\omega_k}\int_0^t f(\tau)\,e^{i\omega_k(t-\tau)}d\tau, \qquad (24)$$

where $f(\tau)$ is the moving load-dependent term and $\Xi_k$ is the effect that the beam's mass and the attached TMD have on the beam's response:

$$\Xi_k = 2\int_0^L \overline{m}(x)[\psi_k(x)]^2 dx + \sum_{j=1}^N TMD_j\left[\psi_k(x_j)\right]^2, \qquad (25)$$

where:

$$TMD_j = \frac{M_{TMD\,j}\left[2\left(k_{TMD\,j} + i\omega_k c_{TMD\,j}\right)^2 - i\omega_k{}^3 M_{TMD\,j}c_{TMD\,j}\right]}{\left[\left(k_{TMD\,j} + i\omega_k c_{TMD\,j}\right) - M_{TMD\,j}\omega_k{}^2\right]^2}. \qquad (26)$$

For a moving load, the response equation takes the following form:

$$w(x,t) = \sum_{k=1}^{\infty}\left(\psi_k(x)\int_0^t e^{i\omega_k(t-\tau)}d\tau\right)\frac{\int_0^L \psi_k(x)\delta(x - V_0\,\tau)\,dx}{i\,\Xi_k\,\omega_k}, \qquad (27)$$

where $\delta(\cdot)$ is a Dirac's delta function and $V_0$ is the velocity of the load.

Further, when considering multiple loads traversing the beam, the effects of preceding loads must also be accounted for [39]:

$$
w(x,t) = \sum_{k=1}^{\infty} \psi_k(x) \, \frac{\int_0^L \psi_k(x)\delta(x - V_0\,\tau)\,dx}{i\,\Xi_k\,\omega_k} \left( \int_{\tau_L^0}^t e^{i\omega_k(t-\tau)}d\tau - \int_{\tau_L^E}^t e^{i\omega_k(t-\tau)}d\tau \right),
\tag{28}
$$

where $\tau_L^0$ and $\tau_L^E$ denote the start and end times of the $L$th load.

Due to the presence of complex conjugate pairs, Equation (28) can revert to the following real form [12]:

$$
w(x,t) = 2\mathrm{Re}\left[ \sum_{k=1}^{\infty} \psi_k(x) \, \frac{\int_0^L \psi_k(x)\delta(x - V_0\,\tau)\,dx}{i\,\Xi_k\,\omega_k} \left( \int_{\tau_L^0}^t e^{i\omega_k(t-\tau)}d\tau - \int_{\tau_L^E}^t e^{i\omega_k(t-\tau)}d\tau \right) \right].
\tag{29}
$$

## 6. Poissonian White Noise Processes

A Poissonian white noise process is a type of delta-correlated process [40], that is "simple, robust and gives accurate results" [41] when used to model loading caused by free flowing traffic, i.e., traffic unencumbered by an unusually heavy volume that causes a continuous series of moving loads due to jams and tailbacks. Poissonian processes are most commonly defined as [11]:

$$
S_P(t) = \sum_{P=1}^{N(t)} Y_P\,\delta(t - T_P),
\tag{30}
$$

where $N(t)$ is a counting function giving the number of impulses in the time interval $[0,t]$, $Y_P$ is the random amplitude of the forcing action, and $\delta(t - T_P)$ is a series of Dirac delta impulses [42] occurring at independent random times $T_P$ following a Poisson type distribution.

When considering a moving load, this characterization of the Poisson type load must be altered, and it is also assumed that the loads will have a constant and equal velocity:

$$
S_P(t) = \sum_{P=1}^{N(t)} Y_P\,\delta[x - (t - T_P)V_0]\;W(t - T_P, t_L),
\tag{31}
$$

where $\delta[x - (t - T_P)V_0]$ is a modification of the Dirac delta function from Equation (30) in which moving loads arriving at random times with random amplitudes are considered, $t_L$ is the time taken for the load to traverse the beam, $L/V_0$ is length divided by the velocity of the moving load, and $W\big(t - t_p, t_L\big)$ is a window function which removes the force after it has traversed the beam; here, $U(\cdot)$ is a unit step function: $W(t - T_P, t_L) = U(\tau)[1 - U(\tau - t_L)]$.

Following the method proposed by [11,43], the Poisson process is filtered to ensure that it is applicable to the beam's characteristics, and this filtering causes Equation (31) to take the following form:

$$
S_k(t) = \sum_{P=1}^{N(t)} Y_P\psi_k(V_0\,\tau)W(t - T_P, t_L).
\tag{32}
$$

Substituting this into the original EoM gives:

$$EI\frac{\widetilde{\partial^4}w(x,t)}{\partial x^4} + \overline{m}\frac{\partial^2 w(x,t)}{\partial t^2} + R(x,t) = \sum_{P=1}^{N(t)} Y_P \psi_k(V_0\, t)\, W(t - T_P, t_L). \tag{33}$$

Considering the theory of separable variable, this can also take the following form in the time domain for an undamped system, which can be readily generalized for damped one [37,44]:

$$\ddot{g}_k(t) + \omega_k^2 g_k(t) = \frac{2}{\Xi_k}\, S_k(t). \tag{34}$$

where $S_k(t)$ is the random forcing action at the $k$th mode, which takes the form:

$$S_k(t) = \sum_{P=1}^{N(t)} Y_P\, \psi_k(V_0\, \tau) W(\tau, t_L). \tag{35}$$

## 7. Validation of the Proposed Method

Prior to presenting the numerical application of this model in which a bridge model fitted with TMDs is subjected to Poisson type loading, this section presents a validation of the proposed method, with reference to existing literature, well known and proven mathematical models, and experimental dynamics.

The method proposed in this paper was developed over a number of years by authors based in the University of Palermo and their national and international co-authors from other institutions. This allowed the accuracy of the numerical method proposed in this paper to be discussed without presenting an extensive new series of numerical analyses. Studies conducted by Failla et al. [33,36,37,44], Di Lorenzo et al. [34,35], and Adam et al. [12] confirmed that the method proposed in this paper is valid by providing exact solutions. When compared to the classical mathematical method, it should be noted that as all mathematical methods are an approximation, so when "exact solutions" are discussed, this term is in reference to the classical solution.

Therefore, considering the classical case of a simply supported single span beam with no intraspan attachments, the following natural frequencies are obtained and displayed in Table 1. The beam in question is 24 m in length, with cross-sectional area of A = 3 m$^2$, a Young's modulus of $E = 215$ GPa, a moment of inertia of $I = 0.25$ m$^4$, a mass per unit length of $\overline{m} = 24{,}150$ Kg, and a density of $\rho = 8050$ Kg/m$^3$. These natural frequencies from the CM were compared with those obtained using the FEM by SAP2000 software [45]. This FEM model used 100 finite elements to provide sufficiently accurate results.

**Table 1.** Comparison of natural modes for a bare beam. FEM: finite element method; CM: classical numerical method.

| Natural Frequency | FEM | CM |
|:---:|:---:|:---:|
| 1 | 25.55841 Rad/s | 25.56277 Rad/s |
| 2 | 102.23362 Rad/s | 102.25108 Rad/s |
| 3 | 230.02565 Rad/s | 230.06494 Rad/s |
| 4 | 408.93450 Rad/s | 409.00434 Rad/s |
| 5 | 638.96015 Rad/s | 639.06927 Rad/s |

As can be seen, there was a very strong agreement between these two sets of results, and, thus, both the classical method and the FEM method could be seen as accurate in this simple case.

As both the classical method and the proposed method use the exact same foundation, the process for solving a bare beam would be identical in both methods. In Tables 2 and 3, comparisons are presented for a simply supported beam with the same parameters fitted with one TMD (Table 2) and two TMDs (Table 3). These show the accuracy of the proposed method (PM) in comparison to the CM and the FEM. Each TMD had the same parameters: a mass ratio of 5% between the concentrated mass and the beam mass, a spring stiffness of $k_{TMD\ j} = 17.17723$ MN/m, and a damping coefficient of the dashpot of $C_{TMD\ j} = 188,522.0486$ Ns/m.

**Table 2.** Comparison of natural modes, 2-span beam. PM: proposed method.

| Natural Frequency | CM | PM | FEM |
|:---:|:---:|:---:|:---:|
| 1 | 21.64388 Rad/s | 21.64388 Rad/s | 21.31504 Rad/s |
| 2 | 102.25108 Rad/s | 102.25108 Rad/s | 102.23362 Rad/s |
| 3 | 230.18479 Rad/s | 230.18479 Rad/s | 230.15534 Rad/s |
| 4 | 409.00434 Rad/s | 409.00434 Rad/s | 408.93449 Rad/s |
| 5 | 639.11179 Rad/s | 639.11179 Rad/s | 639.00635 Rad/s |

**Table 3.** Comparison of natural modes, 3 span beam.

| Natural Frequency | CM | PM | FEM |
|:---:|:---:|:---:|:---:|
| 1 | 20.86973 Rad/s | 20.86973 Rad/s | 20.61325 Rad/s |
| 2 | 102.6735 Rad/s | 102.6735 Rad/s | 102.69047 Rad/s |
| 3 | 230.06494 Rad/s | 230.06494 Rad/s | 230.02566 Rad/s |
| 4 | 409.10419 Rad/s | 409.10419 Rad/s | 409.04306 Rad/s |
| 5 | 639.13335 Rad/s | 639.13335 Rad/s | 639.02946 Rad/s |

In the first of these examples, the TMD was fitted at the midspan, and the results are presented below

As Table 2 shows, there was a very good agreement between the three models, with the proposed method yielding exact results to the classical method.

Table 3 below shows the results of a three-span beam with TMDs fitted at one third of the span and two thirds of the span, 8 and 16 m, respectively; again, the TMDs had the same parameters as previously shown.

Table 3 again shows a very strong agreement between all three mathematical models where, again, the proposed and classical methods had an exact match.

To further attest to the validity of the proposed method, a number of other studies have applied the proposed method, namely [12,33,36,37,39,44,46]; these papers have shown the proposed method's formulation along with results of the dynamic response under various types of forcing actions.

## 8. Numerical Application

In this section, a brief numerical application of the method presented. A bare beam, a beam fitted with three TMDs, and a beam fitted with five TMDs were subjected to Poisson type loading, and their mean and standard deviation responses in terms of midspan displacement are compared.

Figure 1 shows a beam configuration with 5 TMDs with a length of L = 24 m, a cross-sectional area of A = 3 m$^2$, a Young's modulus of $E = 215$ GPa, a moment of inertia of $I = 0.25$ m$^4$, a mass per unit length of $\overline{m} = 24,150$ Kg, and a density of $\rho = 8050$ Kg/m$^3$. The TMD tuning parameters were the same for each TMD and were tuned according to Den Hartog's [47] tuning principles. The mass ratio was first selected as 5%, which gave an ideal frequency ratio of 0.9524 and an optimal damping ratio of 0.1336. From here, the spring stiffness of $k_{TMD\ j} = 17.17723$ MN/m, the dashpot's damping coefficient of $C_{TMD\ j} = 188,522.0486$ Ns/m, and the lumped mass of $M_{TMD\ j} = 28,980$ Kg could be calculated.

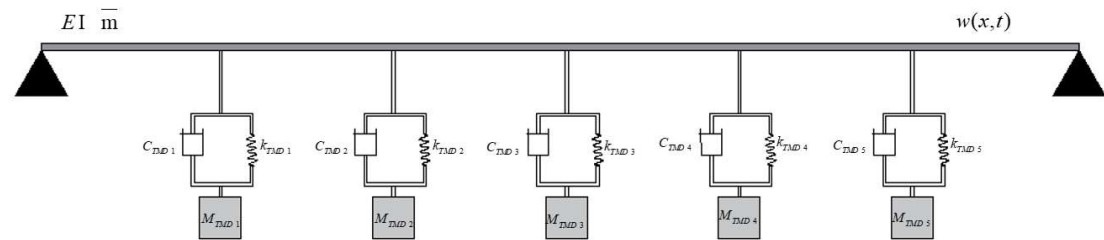

**Figure 1.** Euler–Bernoulli beam fitted with 5 tuned mass dampers (TMDs).

As Table 4 shows, the mathematical method was highly accurate, providing similar results to those obtained by the FEM. There was a rather small margin of error, but this was to be expected with the higher amount of discontinuities considered.

**Table 4.** Comparison of natural frequencies obtained by the FEM and the proposed mathematical method.

|        | FEM             | PM              |
| ------ | --------------- | --------------- |
| Mode 1 | 19.11186 Rad/s  | 19.27554 Rad/s  |
| Mode 2 | 103.14537 Rad/s | 103.09191 Rad/s |
| Mode 3 | 230.41413 Rad/s | 230.42510 Rad/s |
| Mode 4 | 409.15147 Rad/s | 409.20594 Rad/s |
| Mode 5 | 639.09873 Rad/s | 639.19895 Rad/s |

In the following analysis, only the first five modes were considered. As the modal superposition method was applied, the number of modes considered could have had a great effect on the accuracy of the results presented, though it is generally understood that lower modes govern the deflection of a bridge model and can model the response with high accuracy, whereas higher modes contribute, in greater part, to the calculation of bending strain [5].

Considering a beam in which dashpot's were again present, the results from the Poisson type loading are presented. A Monte-Carlo simulation was run 2000 times for each mode—10,000 times in total. The arrival rate set for these loads was 0.375 loads per second, the velocity was 34 m/s, and the magnitude was between the range of 40,000 and 240,000 N.

The figures below show the responses obtained.

Figures 2 and 3 above clearly show that both the standard deviation and the mean displacement of the bridge tended to decrease with an increasing number of TMDs. These results were to be expected, but, nevertheless, the beneficial effect of using multiple dampers to obtain resilient and sustainable structures was shown, as was the accuracy of the proposed method.

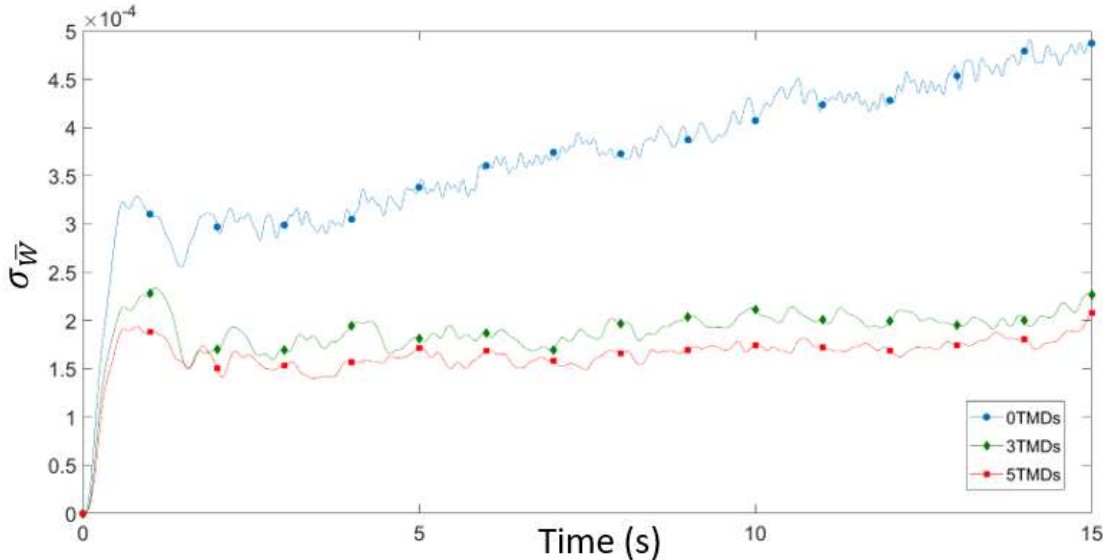

**Figure 2.** Standard deviation of displacement.

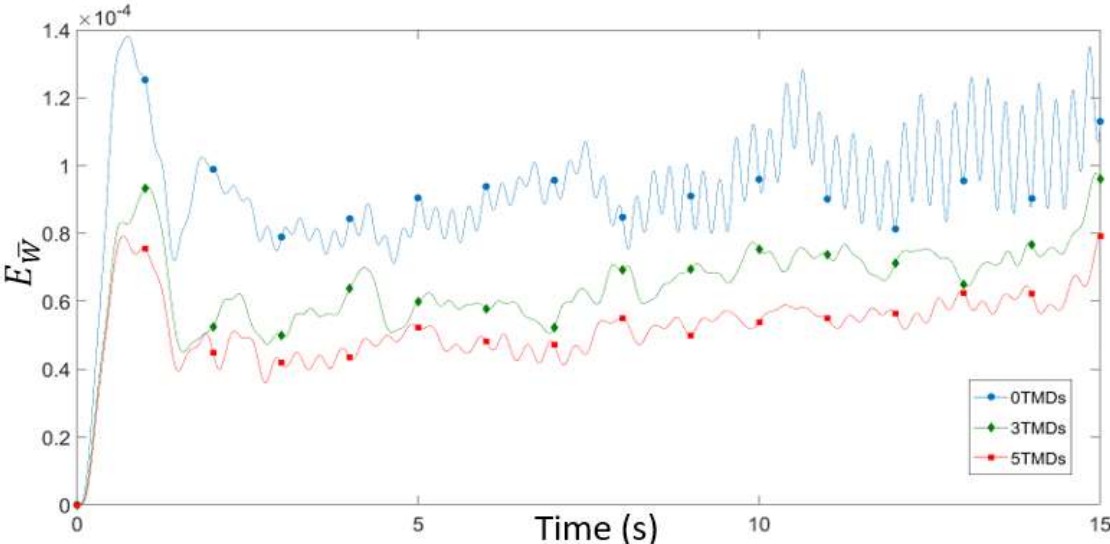

**Figure 3.** Mean displacement.

## 9. Conclusions

A novel method was presented to find the response of an Euler–Bernoulli beam fitted with TMDs subjected to Poisson type loading. This method yields a number of advantages over traditional methods, principally a reduction in the computational power required to find an exact solution. This was achieved by viewing an attachment, in this case a TMD, not as a discontinuity—wherein the beam would have to be split at each attachment point and considered as two separate but coupled continuous beams that are related through their boundary conditions—but rather as a point load. By considering attachments as reactionary forces creating loading at specific points, the number of unknown constants is reduced, which greatly simplifies the classical matrix method solution; when using the proposed method, there will only ever be four unknown constants that must be computed as opposed to the classical case, in which 4(N + 1) constants for N number of attachments must be found. The use of "smart" dampers was also discussed, as these are still relatively new and their application is not yet widespread. This means that this is still a market ripe for expansion, and with more efficient and accurate dynamic analysis, it may be easier to promote the introduction of these novel types of dampers to existing structures. The introduction of these novel types of dampers to the proposed

analytical method was also discussed, and it was then shown that a robust mathematical method can be developed for each type of damper.

This method will hopefully lead to the more widespread application of smart damping devices because it provides an exact, simple, and computationally efficient method to study their effects on rail and road bridges. In the future, it is the authors' aim to extend these applications to specific "smart" technologies that can improve the resilience of transport infrastructures while allowing for the design of more slender structures that use less material, therefore, leading to significant reductions in energy use in the construction phase.

Finally, a Monte-Carlo simulation was run in which three beam configurations were subjected to Poisson type loading. The results were far from surprising, as the bare beam showed the highest displacements, the beam with three TMDs showed a considerable improvement, and the beam with five TMDs showed an even greater improvement—although it could be debated whether or not this extra effort was worth the relatively minor improvement. The method proposed herein was shown to be applicable to cases involving Poisson type loading and can therefore be extended to consider other cases of stochastic loadings and cases involving smart damping devices.

**Author Contributions:** Conceptualization, A.P. and G.F.; methodology, I.P.D.; software, I.P.D. and A.D.M.; validation, A.P., A.D.M. and I.P.D.; formal analysis, I.P.D.; investigation, I.P.D.; resources, A.P.; data curation, I.P.D. and A.D.M.; writing—original draft preparation, I.P.D.; writing—review and editing, I.P.D.; visualization, A.D.M.; supervision, A.P. and G.F.; project administration, A.P.; funding acquisition, A.P. All authors have read and agreed to the published version of the manuscript.

**Funding:** The research presented in this project was carried out as part of the H2020-MSCA-ETN-2016. This project has received funding from the European Union's H2020 Programme for research, technological development and demonstration under grant agreement number 721493.

**Acknowledgments:** Authors gratefully acknowledge the support received from the Italian Ministry of University and Research, through the PRIN 2017 funding scheme (project 2017J4EAYB_002–Multiscale Innovative Materials and Structures "MIMS").

**Conflicts of Interest:** The authors declare no conflict of interest. The funders had no role in the design of the study; in the collection, analyses, or interpretation of data; in the writing of the manuscript, or in the decision to publish the results

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
