# Peer review of "A Novel Solution to Find the Dynamic Response of an Euler–Bernoulli Beam Fitted with Intraspan TMDs under Poisson Type Loading"

_infrastructures, doi:10.3390/infrastructures5050040_

Round 1

Reviewer 1 Report

The subject is interesting, but the paper, in order to be published, requires a thorough revision of the text, both as regards English, which in many cases is very difficult to read, and as regards the formatting of the formulas.
In many cases strange characters appear that do not allow the reviewer to assess the correctness of the formulas. In particular see formulas n. 21, 23, 24 line 264, 25, 26.

The introduction needs to be revised: from line 87 to 94, it promises to deal with a subject (the SMARTI pillars) that is never mentioned in the rest of the text.

Author Response

Thank you for taking the time to review this submission. The authors would like to express their gratitude to you for your invaluable advice and for the constructive criticism which you have offered; we are certain that your contribution will enable us to strengthen the submission.

Regarding your specific points:

“In many cases strange characters appear that do not allow the reviewer to assess the correctness of the formulas. In particular see formulas n. 21, 23, 24 line 264, 25, 26.”

 The strange characters appearing in the equations and inline symbols seems to be a problem with the PDF file. This problem has been resolved.

“The introduction needs to be revised: from line 87 to 94, it promises to deal with a subject (the SMARTI pillars) that is never mentioned in the rest of the text.”

Based on the comments of another reviewer, the authors elected to add a number of paragraphs throughout the paper, within these paragraphs, the SMARTI pillars are discussed. I hope that these paragraphs namely lines  125-126 and 210-218, provide enough information to justify the inclusion of lines 87-94 of the introduction, however, if you judge these additions to be insufficient, we are happy to edit the introduction to remove the references to the SMARTI pillars.

Regarding the more general point on the quality of the English: changes have been made throughout by an author who is a native speaker and we hope that the quality is now sufficient and far less difficult to read.

Thank you once again for your time and effort,

The Authors

Reviewer 2 Report

A Novel Solution to find the Dynamic Response of a
Multi-Span Euler-Bernoulli Beam under Poissonian
Loading

Comments:

  1. The author has proposed a novel approach to obtain the dynamic response of Euler-Bernoulli beam with damping elements such as TMDs. It is just a special type of continuous beam. The title of the manuscript should be modified.
  2. Suggest that Poissonian to be changed to Poison type
  3. In Eq. (15), the author need to explain the meaning of expression , is it a Jacobian matrix?
  4. In section 6, the authors propose to use modal superposition principle to analyze the forced vibration of continuous beams. In numerical calculation, we usually choose finite number of modes for superposition. Therefore, the influence of truncation errors should be considered. In other words, how many modes do we need to consider for a result with enough accuracy.
  5. In the numerical examples, 2000 runs of Monte-Carlo simulation are considered for each model. Is 2000 ensemble enough? How did the authors choose the number ?
  6. The labels for x and y are lacking in Fig.2 and Fig. 3.
  7. numerical application result need to be compared with conventional methods such as FEM to explain it correctly and discuss its advantages.

Author Response

Thank you for taking the time to review this submission. The authors would like to express their gratitude to you for your invaluable advice and for the constructive criticism which you have offered; we are certain that your contribution will enable us to strengthen the submission.

We will attempt to go through each of your points and address them to your satisfaction. Prior to beginning however, I would like to thank you for the clarity with which you reviewed this paper, this made our job of editing and responding to you far easier.

  1. Regarding the title of the manuscript. The title has been edited in the version uploaded and I have contacted the editor to have it revised in the MDPI system, I hope that it is now to your satisfaction.
  2. All instances in which “Poissonian” loading has been discussed have been edited to “Poisson type” There are some notable exceptions to this where the original “Poissonian” has been retained due to references to other works.
  3. A more detail description of the equation presented (eq. 15) has been added, in lines 249-251
  4. You are correct, naturally, that truncation errors are possible by limiting the number of modes considered. In the paper submitted, 5 modes were considered. As the contribution of higher modes is significantly less than the lower modes, the decision to limit our consideration to only the first five modes was made. This is in keeping with a number of similar papers in the field that consider only 4 or 5 modes, more modes could easily be considered using this method but the time that would be taken by the Monte Carlo simulation to consider these modes was deemed prohibitive. In the Numerical application section we have added a very brief discussion on this along with a reference supporting our claim that considering only the first 5 lower modes is sufficiently accurate.
  5. These numbers were chosen rather arbitrarily, 10 000 simulations is a sufficiently large round number and it was our opinion that the results obtained from this were sufficient to illustrate the proposed mathematical method’s applicability. Unfortunately, I cannot provide a better reason than that.
  6. These figures have been edited and now contain x and y labels, please excuse the prior oversight.
  7. In the numerical section, a number of references have been included to attest to the accuracy of this method. In addition, a table has been included comparing the natural frequencies obtained for the first five modes using the finite element method and the mathematical method proposed in the paper. See lines 369-378

Thank you once again for your time, effort, and clarity,

The Authors

Reviewer 3 Report

Authors presents a formulation to compute the dynamic response of a Euler-Bernoulli beam equipped with Tuned Mass Damper. The manuscript deals with the problem only from the mathematical point of view, without proposing a clear application: only a few rows at the end of the paper together with two figures without labels and without units of measurement. In the introduction, authors write that "current approaches used in the design of smart road bridges will be discussed and the importance of robust, efficient, and highly accurate analytical application and validation of these methods will be stressed". This is exactly what should be done, but an evidence of this is almost completely missing at the end of the paper.
The scope of the journal includes applications in road, railway, etc. Moreover, the SMARTI Special Issue requires this application to contain novelty characters related to the following features: sustainable, multi-functional, automated, resilient. Authors speak about sustainable, but reading the application it is very difficult to understand why a beam analysis should contain one of these features.
For all these reason my opinion is that the manuscript is definitely not suitable to be published in the infrastructures journal.

Author Response

Thank you for taking the time to review this submission. The authors would like to express their gratitude to you for your invaluable advice and for the constructive criticism which you have offered; we are certain that your contribution will enable us to strengthen the submission.

We have addressed each of the points that you raised.

Regarding the figures, these have now been repaired, I apologise for the oversight,  we thank you for your attention to detail.

Regarding the suitability,

Although the paper is concerned with theoretical aspects of modelling and analysis of bridge structures, the authors should like to underline that:

Modeling and simulation are becoming increasingly important enablers for the analysis and design of complex systems. In application domains such as structural design, automotive design, mechanical design, biomechanics, transport infrastructure the notion of a "virtual experiment" is a valuable tool for checking and optimizing extensively the complex designs before a realization is ever made.

Virtual experiments, that is analytical models represent a valuable tool to understand the dynamic response of currently existing transport infrastructure, namely rail and road bridges, as well as those currently being designed.

Moreover the proposed analytical approach offers a reliable and effective tool for the design of these systems in several real conditions, for two important points:

An exact solution of the problem is proposed and not an approximate one and this innovative exact solution is obtained without demanding computational time.

Finally, due to the very small computational effort involved in the use of the proposed formulation, and the investigation on its reliability, it is hoped that the proposed approach can be used as preliminary design tool for bridge structural realization. These considerations have been included into the introduction and in the conclusions.

Regarding the SMARTI pillars, in this paper we do mention sustainability, due to the fact that the inclusion of “smart” damping devices in the design of a structure allows more slender structures which use less natural resources and energy to construct. Furthermore, an argument could be made that the implementation of such devices results in more robust structures capable of withstanding higher loading and requiring less maintenance. Again, some lines have been included throughout supporting this argument.

Thank you for your time taken to review this submission and we hope that we will be successful in changing your opinion on the status of the paper,

The Authors.

Round 2

Reviewer 3 Report

My opinion is that the manuscript is still not suitable to be published in the infrastructures journal. Especially in the SMARTI Special Issue that requires the paper contains specific applications with novelty characters.
However, since other two reviewers accepted the current version of the manuscript with only minor revision, I can change my suggestion for the editor, but the authors should make an effort to revise the paper so that the following points are added into the manuscprit:

  1. at least one clear and better explained application in road or railway, etc..;
  2. a complete section dedicated to the validation of the presented method;
  3. an experimental test case. If the authors can not, also a test case based on a detailed FEM model is acceptable.

Authors can combine previous three items using a detailed FEM model, defined in one of the fields of interest for the journal, both for the validation and as test case. But this test case/application must be clearly defined and the FEM model described in detail.
Concerning the comparison of natural frequencies obtained by a FEM and the proposed mathematical method, it makes the doubts on this work grow.
Authors wrote that other articles already in the literature confirm the method proposed in this paper is valid providing exact solutions when compared to the classical mathematical method.
Does this mean that the proposed method is not new?
The comparison shows a error of the 15% on the frequency of first mode. This is not a negligible error. If the presented method provide exact solutions, like authors said, maybe the FEM model is not suitable for this validation and a different model must be selected. Maybe a more detailed FEM model. This different model could be the test case/application I mentioned above.
However, any new method presented in literature needs a rigorous validation campaign.
The sentence 'FEM, which, must be stressed, does not provide exact results.' should be removed. FEM simulations represent an approximation, but every mathematical model represents an approximation of the reality, even what the authors call an exact solution. Moreover, the FEM approximation cannot justify a 15% error.
What might be interesting is the comparison between the compuational time of the proposed method against the FEM model. Authors speak about this aspect but do not provide any data.
In general, the manuscprit must be improved a lot in the validation, results and conclusions sections.

Author Response

We would like to begin by thanking you for your willingness to change your opinion regarding the suitability of the article for the infrastructures journal. We understand that the content of this paper, while applicable to the testing and analysis of rail and road bridges, does not propose or even attempt to develop a novel device which can be implemented on the aforementioned structures. We understand and can agree with this perspective however, we feel that virtual testing through accurate modelling can provide vast benefits in the design of future infrastructure which does fit with the scope of the journal despite not suggesting a prototype.

1 at least one clear and better explained application in road or railway, etc..;

2 a complete section dedicated to the validation of the presented method;

3 an experimental test case. If the authors can not, also a test case based on a detailed FEM model is acceptable.

These points are contained in the validation section

“Authors can combine previous three items using a detailed FEM model, defined in one of the fields of interest for the journal, both for the validation and as test case. But this test case/application must be clearly defined and the FEM model described in detail.
Concerning the comparison of natural frequencies obtained by a FEM and the proposed mathematical method, it makes the doubts on this work grow.”

We trust that the new FEM model and results will eliminate these doubts as the results are far more accurate.

“Authors wrote that other articles already in the literature confirm the method proposed in this paper is valid providing exact solutions when compared to the classical mathematical method.
Does this mean that the proposed method is not new?”

This is correct, the method proposed in this paper has went through a number of iterations in its development. What is presented here is an extension to the proposed method, all of the previous developments are cited throughout. References [12,33,34,36,37,44,45]

The comparison shows a error of the 15% on the frequency of first mode. This is not a negligible error. If the presented method provide exact solutions, like authors said, maybe the FEM model is not suitable for this validation and a different model must be selected. Maybe a more detailed FEM model. This different model could be the test case/application I mentioned above.

This is in all likelihood the correct conclusion. A different comparison has been presented using different FEM software, this provides a far better correlation. In all probability there was an underlying issue in the larger scale FEM model due to the effects of shear effects which are not included in the Euler-Bernoulli model. An FEM model was produced which does not consider shear effects and these results provide a markedly better correlation

However, any new method presented in literature needs a rigorous validation campaign.
The sentence 'FEM, which, must be stressed, does not provide exact results.' should be removed. FEM simulations represent an approximation, but every mathematical model represents an approximation of the reality, even what the authors call an exact solution. Moreover, the FEM approximation cannot justify a 15% error.

This sentence has been removed and, as previously stated, a more favourable comparison with far better correlation between the two methods has been presented. Regarding the use of the term “exact” this is in reference to the classical method, the Proposed method has previously been validated in comparison to the classical method and a further evaluation has been shown in the revised paper.

What might be interesting is the comparison between the compuational time of the proposed method against the FEM model. Authors speak about this aspect but do not provide any data.

This is something that is difficult to include due to the varying computing power available on each system. What I can tell you is that the difference on my laptop between the proposed and classical methods for a 6 span beam equipped with 5 TMDs is vast. With the proposed method, this analysis takes under 1 minute whereas the same model with the classical method takes over an hour. It is possible however that with a more powerful computer and with a more efficient script, these results would vary drastically, as such, we have elected to omit exact information for fear that it would appear misleading.

In general, the manuscript must be improved a lot in the validation, results and conclusions sections.

These sections have been updated based on your suggestions and we trust that you will find these changes to be satisfactory.

Best regards,

The Authors
